# Comparative Efficacy and Safety of Lorlatinib and Alectinib for ALK-Rearrangement Positive Advanced Non-Small Cell Lung Cancer in Asian and Non-Asian Patients: A Systematic Review and Network Meta-Analysis

**DOI:** 10.3390/cancers13153704

**Published:** 2021-07-23

**Authors:** Koichi Ando, Ryo Manabe, Yasunari Kishino, Sojiro Kusumoto, Toshimitsu Yamaoka, Akihiko Tanaka, Tohru Ohmori, Hironori Sagara

**Affiliations:** 1Division of Respiratory Medicine and Allergology, Department of Medicine, Showa University School of Medicine, 1-5-8 Hatanodai, Shinagawa-ku, Tokyo 142-8666, Japan; r.manabe@med.showa-u.ac.jp (R.M.); ookiyookiy@med.showa-u.ac.jp (Y.K.); k-sojiro@med.showa-u.ac.jp (S.K.); tanakaa@med.showa-u.ac.jp (A.T.); ohmorit@med.showa-u.ac.jp (T.O.); sagarah@med.showa-u.ac.jp (H.S.); 2Division of Internal Medicine, Showa University Dental Hospital Medical Clinic, Senzoku Campus, Showa University, 2-1-1 Kita-senzoku, Ohta-ku, Tokyo 145-8515, Japan; 3Advanced Cancer Translational Research Institute, Showa University, 1-5-8 Hatanodai, Shinagawa-ku, Tokyo 142-8555, Japan; yamaoka.t@med.showa-u.ac.jp; 4Department of Medicine, Division of Respiratory Medicine, Tokyo Metropolitan Health and Hospitals Corporation, Ebara Hospital, 4-5-10 Higashiyukigaya, Ohta-ku, Tokyo 145-0065, Japan

**Keywords:** alectinib, ALK rearrangement, lorlatinib

## Abstract

**Simple Summary:**

The treatment of anaplastic lymphoma kinase (ALK) rearrangement-positive (ALK-p) advanced non-small cell lung cancer (NSCLC) remains a challenge. We compared the safety and efficacy of lorlatinib and alectinib in patients with ALK-p ALK-inhibitor‒naïve advanced NSCLC (in overall participants and in the Asian and non-Asian subgroups). The results showed that in the overall participant group, the efficacy of lorlatinib and alectinib was not significantly different in terms of progression-free survival (PFS) and overall survival (OS). Although in the Asian subgroup, PFS was not significantly different upon treatment with lorlatinib or alectinib, in the non-Asian subgroup, PFS was significantly better in response to lorlatinib than with alectinib. Grade 3 or higher adverse events in the overall participant group were significantly more frequent with lorlatinib than with alectinib. These results will provide valuable information that would enable the improvement of treatment strategies for ALK-p ALK-inhibitor‒naïve advanced NSCLC.

**Abstract:**

To date, there have been no head-to-head randomized controlled trials (RCTs) comparing the safety and efficacy of lorlatinib and alectinib in anaplastic lymphoma kinase (ALK) rearrangement-positive (ALK-p) ALK-inhibitor‒naïve advanced non-small cell lung cancer (NSCLC). We performed a network meta-analysis comparing six treatment arms (lorlatinib, brigatinib, alectinib, ceritinib, crizotinib, and platinum-based chemotherapy) in overall participants and in Asian and non-Asian subgroups. Primary endpoints were progression-free survival (PFS), overall survival (OS), and grade 3 or higher adverse events (G3-AEs). There were no significant differences between lorlatinib and alectinib in overall participants for both PFS (hazard ratio [HR], 0.742; 95% credible interval [CrI], 0.466–1.180) and OS (HR, 1.180; 95% CrI, 0.590–2.354). In the Asian subgroup, there were no significant differences in PFS between lorlatinib and alectinib (HR, 1.423; 95% CrI, 0.748–2.708); however, in the non-Asian subgroup, PFS was significantly better with lorlatinib than with alectinib (HR, 0.388; 95% CrI, 0.195–0.769). The incidence of G3-AEs in overall participants was significantly higher with lorlatinib than with alectinib (risk ratio, 1.918; 95% CrI, 1.486–2.475). These results provide valuable information regarding the safety and efficacy of lorlatinib in ALK-p ALK-inhibitor‒naïve advanced NSCLC. Larger head-to-head RCTs are needed to validate the study results.

## 1. Introduction

Studies conducted in the field of molecular biology over the past decade have contributed substantially to a better understanding of the molecular mechanisms underlying the development and progression of non-small cell lung cancer (NSCLC) [1,2]. This has enabled remarkable progress in the development of treatment strategies for NSCLC, which accounts for 84% of all lung cancers [1,2,3]. Despite recent improvements in treatment strategies, lung cancer remains the leading cause of cancer-related deaths globally, accounting for 13% of all cancer-related deaths [4,5]. The overall five-year survival rate of localized lung cancer is 63%. However, most NSCLCs are diagnosed at an advanced stage, and the five-year survival rate for metastatic lung cancer—even with systemic anti-cancer treatment—is only 7% [1,2,4,5,6,7]. There has always been a need for the constant improvement and development of therapeutic strategies.

Importantly, 3–5% of the patients with NSCLC harbor anaplastic lymphoma kinase (ALK) gene fusions, with the most common gene fusion being echinoderm microtubule-associated protein-like 4-ALK (EML4-ALK) [1,8,9,10,11]. EML4-ALK activates the phosphatidylinositol 3-kinase-AKT (PI3K-AKT), reticular activating system (RAS), and Janus kinase/signal transducer and activator of transcription (JAK/STAT) signaling cascades, which affect tumor progression, survival, and growth, and by extension other organ metastases, such as the central nervous system (CNS) metastases [10,11]. ALK inhibitors, such as crizotinib, alectinib, brigatinib, and lorlatinib can act on mutated ALK and inhibit the production of ALK fusion proteins resulting from ALK rearrangements [1,10,12,13,14] (Figure 1).

Lung cancer is associated with a high rate of central nervous system (CNS) metastases [15,16,17]. ALK rearrangement-positive (ALK-p) NSCLC has been reported to exhibit a high frequency of CNS metastases, which is associated with worse prognosis and reduced quality of life (QOL) [17,18]. As NSCLC accounts for more than 84% of all lung cancers, the treatment of ALK-p NSCLC with CNS metastases is becoming increasingly important as advances in lung cancer treatment ensure prolonged survival [5,15,17]. However, specific therapeutic strategies for ALK-p NSCLC with CNS metastases have not yet been fully established.

Alectinib, an ALK inhibitor has a benzo[b]carbazole backbone and is highly selective for ALK in contrast to other ALK inhibitors, such as crizotinib and ceritinib [19,20,21]. Results from the ALEX and J-ALEX phase III studies on alectinib showed that alectinib significantly prolonged the progression-free survival (PFS) (compared to crizotinib), and that the incidence of Grade 3 or higher adverse events was lower with alectinib than that with crizotinib [22,23]. Based on these results, the United States (US) National Comprehensive Cancer Network (NCCN) guidelines and the Japanese lung cancer practice guidelines recommend alectinib as the first-line treatment for ALK-p advanced NSCLC [3,24]. However, even in patients who respond to alectinib, resistance develops within a few years of treatment [19,20,25]. Resistance to alectinib has been shown to be caused by mutations in the ALK gene locus that result in changes in the conformation of the ALK fusion protein [19,20,26].

Lorlatinib, a third-generation ALK inhibitor, was approved by the US food and drug administration (FDA) as an orphan drug for NSCLC in 2015 and as the second- or third-line treatment for ALK-p advanced NSCLC in 2018 [27]. In Japan, lorlatinib became available in 2018 for use in ALK-p advanced NSCLC that had relapsed or were refractory to ALK inhibitors [24]. It has been reported that lorlatinib is effective in cancers harboring mutations in ALK that conferred resistance to ALK inhibitors, including G1202R and I1171N mutations [28,29]. A remarkable feature of lorlatinib is that it is designed to cross the blood–brain barrier. This makes it highly effective for the treatment of ALK-p advanced NSCLCs, even those with CNS metastases [16,28,29,30]. Several in vivo studies have reported that lorlatinib reduces the size of NSCLC brain metastases in mouse models in a concentration-dependent manner [31,32]. A recent phase III trial has reported that patients with ALK-p ALK-inhibitor-naïve advanced NSCLC (with and without CNS metastases) who had been treated with lorlatinib showed a prolonged PFS compared to those treated with crizotinib [33].

However, to date, there have been no large randomized controlled trials (RCTs) comparing the efficacy of lorlatinib with that of alectinib, currently positioned as first-line therapy in patients with ALK-inhibitor-naïve NSCLC. It is important to compare the safety and efficacy of lorlatinib with those of alectinib in patients with ALK-p ALK-inhibitor-naïve NSCLC to develop effective treatment strategies that can improve patient prognosis. In addition, validation in subgroups with and without CNS metastases would provide additional valuable information.

Notably, previous clinical trials investigating lorlatinib have indicated that race is an important determinant of the efficacy of lorlatinib, i.e., lorlatinib exhibited better efficacy in non-Asians than in Asians. As the results of the comparative analyses of the efficacy and safety profiles of lorlatinib and alectinib may be discrepant owing to the race factor, there is a need for race-specific comparisons of lorlatinib and alectinib.

In addition, a phase III trial of lorlatinib showed that the incidence of Grade 3 or higher adverse events was more frequent in the lorlatinib arm than that in the crizotinib arm. Based on these results, we believe that there is a need to examine the safety profiles of lorlatinib and alectinib in greater detail, i.e., to compare the safety of lorlatinib with that of alectinib in terms of representative safety outcomes, including the incidence of any adverse events (AAEs), serious adverse events (SAEs), increased AST or ALT levels, nausea, diarrhea, and pneumonitis that are relatively ubiquitous with the use of ALK inhibitors and/or often influence the treatment course and prognosis.

A comparison of lorlatinib not only with alectinib but also with other existing ALK inhibitors, including brigatinib, ceritinib, and crizotinib, may also provide valuable information. In addition, analyses that consider not just race or the presence or absence of CNS metastases but also the performance status (PS) may also provide useful information that can be leveraged to develop and improve treatment strategies.

Although RCTs represent the most suitable method for comparing the safety and efficacy of therapies, conducting a large-scale RCT requires considerable time, effort, and significant implementation costs. Therefore, in this study, we adopted the statistical method of network meta-analysis (NMA) to compare the drugs with respect to safety and efficacy [34,35,36,37,38,39]. This study design allows us to compare and even rank the safety and efficacy of any treatment pair, even if there have been no prior RCTs that involved the reporting of direct comparisons [34,35,36].

Therefore, in this systematic review and NMA (registration: UMIN-CTR no. UMIN000043994), we compared and ranked the efficacy and safety of six treatment groups, including lorlatinib, brigatinib, alectinib, ceritinib, crizotinib, and platinum-based chemotherapy, in patients with ALK-p advanced NSCLC, using the Bayesian NMA statistical method [37,40,41]. In addition, we performed subgroup analyses based on race (Asian or non-Asian), the presence or absence of CNS metastases, and PS.

The primary aim of this systematic review and NMA was to compare the safety and efficacy of lorlatinib and alectinib in patients with ALK-p ALK-inhibitor-naïve and advanced NSCLC (in the overall group of participants and in subgroups comprising Asian and non-Asian patients).

## 2. Materials and Methods

### 2.1. Systematic Review

A comprehensive literature search was conducted to identify relevant published reports from 1946 to date. On 6 May 2021, four databases (PubMed [42], Cochrane Library [43], EMBASE [44], and SCOPUS [45]) were searched. The search strategy was developed using keywords, such as NSCLC, ALK inhibitors, lorlatinib, brigatinib, alectinib, ceritinib, crizotinib, and their medical subject headings (MeSH) terms. The search formulae used in PubMed are presented in Appendix B. In addition, to ensure certainty, we checked the reference lists of the retrieved studies to avoid missing relevant studies that met the inclusion criteria. If sufficient information or data could not be found in the literature, the corresponding author was contacted by e-mail, if necessary. The EMBASE, CENTRAL, and SCOPUS databases were also searched using the search strategy used in the PubMed search. The main objective of this systematic review was to identify all the published RCTs in order to compare and rank the safety and efficacy of the six treatment arms, i.e., lorlatinib, brigatinib, alectinib, ceritinib, crizotinib, and chemotherapy in patients with ALK-p advanced NSCLC. To identify all the relevant studies and minimize publication bias, a manual search of the relevant articles was conducted in addition to a review of the references presented in the article. This study was conducted in accordance with the preferred reporting items for systematic review and meta-analysis (PRISMA) guidelines [46] and the preferred reporting items for systematic review and meta-analysis (PRISMA) extended guidelines [35,47]. Two investigators (KA and YK) independently conducted the literature search. To address any clinical or methodological heterogeneity between studies and to ensure the validity of the indirect comparative analysis, inclusion and exclusion criteria were adopted using the PICOS (patients, interventions, comparison, outcomes, and study design) approach for the retrieved studies.

### 2.2. Quality Evaluation

We assessed the quality of the RCTs included in the NMA using the risk of bias tool 2 (RoB2) recommended by the Cochrane Collaboration [48]. The following parameters were rated either as low risk, of some concern, or high risk: (1) Bias arising from the randomization process; (2) Bias due to deviations from intended interventions; (3) Bias due to missing outcome data; (4) Bias in measurement of the outcome; and (5) Bias in selection of the reported result. Evaluations were performed independently by two researchers (KA and SK), and any conflicts were resolved by the third researcher (TY).

### 2.3. Inclusion and Exclusion Criteria (Predefined PICOS)

#### 2.3.1. Patients

The following were the inclusion criteria considered for this study: (1) a minimum age of 18 years; (2) histological or cytological confirmation of advanced or metastatic ALK-p NSCLC, with at least one measurable lesion evaluated in accordance with the response evaluation criteria in solid tumors (RECIST version 1.1.25), with a performance status of 0–2 (on a five-point scale, with higher numbers reflecting greater disability); and (3) no previous exposure to ALK-targeted therapy.

#### 2.3.2. Interventions and Comparisons

For this analysis, we considered oral lorlatinib (dose, 100 mg daily), brigatinib (dose, 180 mg once daily), oral alectinib (dose, 300 or 600 mg twice daily), oral ceritinib (dose, 750 mg daily), oral crizotinib (dose, 250 mg twice daily), and platinum-based chemotherapy, all of which were dosages and modes of administration that were licensed, recommended, or specified in phase III studies. Studies involving either of these treatment arms were eligible in our analysis. We considered crizotinib or platinum–based chemotherapy as the common comparator for each therapeutic agent. This was because, prior to the approval of alectinib and lorlatinib, crizotinib was the first choice in the initial treatment, and prior to the approval of crizotinib, platinum-based chemotherapy was the first-line treatment for ALK-p treatment-naive NSCLC.

#### 2.3.3. Outcomes

The primary efficacy endpoints were PFS and overall survival (OS)—which are common efficacy endpoints in clinical oncology—in the overall participant group and in the Asian and non-Asian subgroups; the corresponding hazard ratios (HRs) and 95% credible intervals (CrIs) were calculated. The secondary efficacy endpoints were PFS and OS in patients with and without CNS metastases and patients with PS of 0–1; the corresponding HRs and 95% CrIs were calculated. The proportion of patients with objective responses (ObRs), which were defined as complete or partial response, was also predefined as the secondary efficacy endpoint; the corresponding odds ratios (ORs) and 95% CrIs were calculated. The primary safety endpoint was the incidence of Grade 3 or higher adverse events (G3-AEs), and the corresponding risk ratios (RRs) and 95% CrIs were calculated. As secondary safety endpoints, the incidence of any grade of any adverse events (AG-AEs), any grade and Grade 3 or higher serious adverse events (AG-SAEs and G3-SAEs), increased-AST levels (AG-AST and G3-AST), increased ALT levels (AG-ALT and G3-ALT), nausea (AG-nausea and G3-nausea), diarrhea (AG-diarrhea and G3-diarrhea), and pneumonitis (AG-pneumonitis and G3-pneumonitis) were examined, and the corresponding risk ratios (RRs) and 95% CrIs were calculated. To rank the safety and efficacy of both treatments, the surface under the cumulative ranking curve (SUCRA) values were calculated for each outcome. A higher SUCRA indicated more favorable treatment in terms of the corresponding endpoints. To be included in this systematic review and NMA, at least one predetermined efficacy or safety endpoint had to be included in the trial under analysis. These defined endpoints were analyzed only when data were available from the included studies; two authors (KA and TY) extracted the relevant data independently and consulted the third author (TO) to resolve any discrepancies when necessary.

#### 2.3.4. Study Design

The studies that were included in this systematic review and meta-analysis were phase III trials of parallel-group RCTs.

### 2.4. Statistical Analysis Method of Network Meta-Analysis

We performed the Bayesian NMA in accordance with a robustly established method that was developed by the National Center for Medical Research [49,50]. We employed the standard Bayesian model described by Dias et al. [51,52,53], which presupposes inconsistency and heterogeneity among the included studies. A non-informative prior distribution was applied, and Gibbs sampling was used to evaluate the posterior distribution of the effect size based on the Markov chain Monte Carlo method [37,41]. The number of iterations was set to 50,000, and the first 10,000 iterations were considered burn-in samples to eliminate the effect of the initial values. The effect sizes were represented by HRs, ORs, and RRs and their 95% CrIs; the difference in the effect size of each endpoint between the treatment groups was considered to be significant if the 95% CrI did not include 1. The SUCRA value varied from 0% to 100%, with a higher SUCRA value indicating a better treatment outcome [54]. A convergent diagnosis was also conducted for all comparisons using the Brooks–Gelman–Rubin (BGR) diagnostic method [55,56]. Both visual and BGR diagnostics were used to ascertain the convergence of the models. OpenBUGS 1.4.0 (MRC Biostatistics Unit, Cambridge Public Health Research Institute, Cambridge, UK) was used for the analysis, and STATA (ver. 14, StataCorp., College Station, TX, USA) was used to visualize the results (College Station, TX, USA).

### 2.5. Sensitivity Analysis

Based on the presence of any conceptual heterogeneity among the included studies, we performed a sensitivity analysis [57,58] by excluding studies that were deemed to be heterogeneous. This made it possible to evaluate whether the inclusion or exclusion of conceptually heterogeneous studies would affect the final conclusions.

### 2.6. Assessment for Between-Study Heterogeneity

Considering the possibility that between-study heterogeneity might affect the final conclusions, we assessed the statistical between-study heterogeneity. Statistical between-study heterogeneity was expressed as the *I*^2^ statistic (%) [59]. We determined that a high degree of heterogeneity existed between studies when the *I*^2^ statistic was greater than 50% [59]. To calculate the *I*^2^ statistic, we performed a pairwise meta-analysis using a random-effects model among the included studies with the same direct comparison.

### 2.7. Ethical Aspects

Institutional review board approval and patient consent were exempted due to the nature of this systematic review.

## 3. Results

### 3.1. Systematic Review

Our systematic literature review identified 1480 studies (361 from PubMed, 259 from EMBASE, 65 from CENTRAL, and 795 from SCOPUS), and 1051 articles remained after removal of duplicates. After adopting the PICOS approach, a total of eight studies were selected for the NMA analysis; two studies that compared crizotinib with chemotherapy (PROFILE1014 [60], PROFILE1029 [61]), three studies that compared alectinib with crizotinib (ALEX [22], J-ALEX [23], and ALESIA [62]), and one study each that compared ceritinib with chemotherapy, brigatinib with crizotinib, and lorlatinib with crizotinib (ACEND-4 [63], ALTA-1L [64], and CROWN [33], respectively). The process employed for the selection of the studies is shown in Figure 2, the main inclusion criteria for each of the included studies are shown in Appendix A, and the main characteristics of the included studies are shown in Appendix A. The analysis was performed on the data of all 2194 patients (corresponding to eight studies; chemotherapy: 461, crizotinib: 878, ceritinib: 189, alectinib: 380, brigatinib: 137, lorlatinib: 149). The network map of this analysis is shown in Figure 3.

### 3.2. Primary Efficacy Endpoint: PFS

There were no significant differences in the PFS of individuals treated with lorlatinib and alectinib (HR, 0.742; 95% CrI, 0.466–1.180). However, PFS was significantly superior in patients treated with lorlatinib than with chemotherapy (HR, 0.121; 95% CrI, 0.078–0.187), crizotinib (HR, 0.280; 95% CrI, 0.191–0.411), ceritinib (HR, 0.220; 95% CrI, 0.131–0.367), or brigatinib (HR, 0.572; 95% CrI, 0.326–0.997) (Figure 4).

The results of the comparison of PFS among patients treated with each pair of the six therapies (including the comparison between lorlatinib and the other therapeutic agents shown in Figure 4) are presented in Appendix A.

Ranking assessment using SUCRA values revealed that patients treated with lorlatinib (SUCRA = 97.4%) had the highest PFS, followed by those treated with alectinib (SUCRA = 79.2%), brigatinib (SUCRA = 63.4%), crizotinib (SUCRA = 38.3%), ceritinib (SUCRA = 21.7%), and chemotherapy (SUCRA = 0.0%) (Appendix A).

### 3.3. Subgroup Analysis for the Primary Efficacy Endpoint (PFS) in the Non-Asian and Asian Subgroups

Only five studies (PROFILE1014, ASCEND-4, ALEX, ALTA-1L, and CROWN) were included for analyzing the non-Asian patients (three studies [PROFILE 1029, J-ALEX, and ALESIA] were conducted only on Asian patients). All eight studies were included for analyzing the Asian patients.

#### 3.3.1. Subgroup Analysis for the Primary Efficacy Endpoint (PFS) in the Non-Asian Subgroup

In the non-Asian subgroup, PFS was significantly higher in lorlatinib-treated patients than in alectinib-treated patients (HR, 0.388; 95% CrI, 0.195–0.769). PFS was also significantly higher in lorlatinib-treated patients than in patients treated with each of the other agents, including chemotherapy (HR, 0.101; 95% CrI, 0.052–0.193), crizotinib (HR, 0.190; 95% CrI, (0.112–0.324), ceritinib (HR, 0.229; 95% CrI, 0.107–0.489), and brigatinib (HR, 0.352; 95% CrI, 0.169–0.732) (Figure 5a). The results of the PFS comparison in patients treated with each pair of the six therapies, including the comparison between lorlatinib and the other therapeutic agents in non-Asian patients, are shown in Appendix A. The results of the ranking assessment in the non-Asian subgroup using SUCRA values showed that patients treated with lorlatinib (SUCRA = 99.9%) had the highest PFS, followed by those treated with alectinib (SUCRA = 71.0%), brigatinib (SUCRA = 65.1%), ceritinib (SUCRA = 38.8%), crizotinib (SUCRA = 25.2%), and chemotherapy (SUCRA = 0.0%) (Appendix A).

#### 3.3.2. Subgroup Analysis for the Primary Efficacy Endpoint (PFS) in the Asian Subgroup

In the Asian subgroup, there were no significant differences in the PFS of patients treated with lorlatinib and alectinib (HR, 1.423; 95% CrI, 0.748–2.708) and lorlatinib and brigatinib (HR, 1.148; 95% CrI, 0.456–2.860). However, PFS was significantly higher in lorlatinib treated patients than that in chemotherapy-treated patients (HR, 0.196; 95% CrI, 0.107–0.363), crizotinib-treated patients (HR, 0.471; 95% CrI, 0.270–0.818), and ceritinib-treated patients (HR, 0.298; 95% CrI, 0.137–0.643) (Figure 5b). The results of the comparison of PFS in patients treated with each pair of the six therapies, including comparison between lorlatinib and other therapeutics in the Asian subgroup are shown in Appendix A. The results of the ranking assessment in the Asian subgroup based on SUCRA values showed that patients treated with alectinib (SUCRA = 91.2%) had the highest PFS, followed by those treated with brigatinib (SUCRA = 78.1%), lorlatinib (SUCRA = 70.4%), crizotinib (SUCRA = 39.3%), ceritinib (SUCRA = 20.1%), and chemotherapy (SUCRA = 0.9%) (Appendix A).

### 3.4. Subgroup Analysis for the Primary Efficacy Endpoint (PFS) in the Subgroups with and without CNS Metastases

Seven studies (PROFILE1014, PROFILE1029, ASCEND-4, ALEX, J-ALEX, ALTA-1L, and CROWN) were included to analyze patient subgroups with and without CNS metastases. In the ALESIA study, PFS was analyzed according to two methodologies: investigator-assessed and independent review committee-assessed PFS. The primary endpoint was investigator-assessed PFS and a secondary endpoint was independent review committee-assessed PFS. In our present analysis, PFS in overall participants included the results of the primary endpoint, investigator-assessed PFS. However, the subgroup analysis of CNS metastasis in ALESIA reported only the results of independent review committee-assessed PFS, and the results of the subset analysis of CNS metastasis for investigator-assessed PFS were not reported. Therefore, considering the discrepancy between investigator-assessed and independent review committee-assessed PFS in the ALESIA study, ALESIA could not be included in the subgroup analysis of CNS metastasis in our present network meta-analysis.

#### 3.4.1. Subgroup Analysis for the Primary Efficacy Endpoint (PFS) in the Subgroups with CNS Metastasis

In the subgroup with CNS metastases, no significant differences were observed between the PFS of patients treated with lorlatinib and alectinib (HR, 0.542; 95% CrI, 0.229–1.285). However, PFS was significantly higher in patients treated with lorlatinib than that in patients treated with chemotherapy (HR, 0.108; 95% CrI, 0.047–0.248), crizotinib (HR, 0.200; 95% CrI, 0.097–0.414), and ceritinib (HR, 0.155; 95% CrI, 0.060–0.398). There were no significant differences in PFS between patients treated with lorlatinib and brigatinib (HR, 1.003; 95% CrI, 0.333–2.979) (Figure 6a). The results of the PFS comparison in patients treated with each pair of the six therapies, including the comparison between lorlatinib and the other therapeutic agents in the subgroup with CNS metastases are shown in Appendix A. Ranking assessment in subgroups with CNS metastasis using SUCRA values showed that patients treated with lorlatinib (SUCRA = 88.3%) had the highest PFS, followed by those treated with brigatinib (SUCRA = 88.0%), alectinib (SUCRA = 63.6%), crizotinib (SUCRA = 35.9%), ceritinib (SUCRA = 22.7%), and chemotherapy (SUCRA = 1.4%) (Appendix A).

#### 3.4.2. Subgroup Analysis for the Primary Efficacy Endpoint (PFS) in the Subgroup without CNS Metastasis

In the subgroup without CNS metastasis, there were no significant differences in PFS between patients treated with lorlatinib and alectinib (HR, 0.705; 95% CrI, 0.402–1.234). However, PFS was significantly higher in patients treated with lorlatinib than the patients treated with chemotherapy (HR, 0.135; 95% CrI, 0.081–0.226), crizotinib (HR, 0.320; 95% CrI, 0.205–0.501), ceritinib (HR, 0.283; 95% CrI, 0.152–0.523), and brigatinib (HR, 0.445; 95% CrI, 0.227–0.864) (Figure 6b). The results of PFS comparison between patients treated with each pair of the six therapies, including comparison between lorlatinib and the other therapeutic agents in the subgroups without CNS metastases, are shown in Appendix A. Ranking assessment in the subgroups without CNS metastasis using SUCRA values showed that patients treated with lorlatinib (SUCRA = 97.6%) had the highest PFS, followed by those treated with alectinib (SUCRA = 80.9%), brigatinib (SUCRA = 57.9%), crizotinib (SUCRA = 36.2%), ceritinib (SUCRA = 27.4%), and chemotherapy (SUCRA = 0.0%) (Appendix A).

### 3.5. Subgroup Analysis by PS for PFS

We attempted to perform subgroup analyses based on the PS by dividing the patient subgroups into individuals with a PS of 0–1 and those with a PS of 2. However, the study of lorlatinib did not provide data of patients in the subgroup with a PS of 2; thus, we were unable to compare the efficacy in the patient subgroup with a PS of 2 between lorlatinib and other existing treatments. Therefore, we performed a subgroup analysis of patients in the subgroup with a PS of 0–1. In addition, we were unable to include ceritinib in the comparison group because the ASCEND-4 study did not report the proportion of patients with a PS of 2. Finally, subset analysis by PS was performed only in the group of patients with PS 0–1 among five treatment groups: lorlatinib, brigatinib, alectinib, crizotinib, and chemotherapy. Seven studies (PROFILE1014, PROFILE1029, ALEX, J-ALEX, ALESIA, ALTA-1L, and CROWN) were included in the analysis of patients’ subgroup with a PS of 0–1. The ASCEND-4 study could not be included due to the missing data required for this subgroup analysis.

In the subgroups of patients with a PS of 0–1, there were no significant differences in PFS between lorlatinib and alectinib (HR, 0.774; 95% CrI 0.486–1.233). However, PFS was significantly superior in the lorlatinib-treated group than in the chemotherapy-treated (HR, 0.121; 95% CrI, 0.077–0.190) and crizotinib-treated (HR, 0.280; 95% CrI, 0.188–0.416) groups. There were no significant differences in PFS between lorlatinib and brigatinib (HR, 0.562; 95% CrI, 0.310–1.025). The results of the PFS comparison between each pair of the five therapies in the patient subgroup with PS of 0–1 are shown in Appendix A. The results of the ranking assessment in the patient subgroups with a PS of 0–1 using the SUCRA value showed that treatment with lorlatinib (SUCRA = 95.7%) had the highest PFS, followed by alectinib (SUCRA = 75.8%), brigatinib (SUCRA = 53.5%), crizotinib (SUCRA = 25.0%), and chemotherapy (SUCRA = 0.0%) (Appendix A).

### 3.6. Primary Efficacy Endpoint: OS

Six studies (PROFILE1014, ASCEND-4, ALEX, ALESIA, ALTA-1L, and CROWN) were included in the analysis for OS. There were no significant differences in OS not only between lorlatinib and alectinib (HR, 1.180; 95% CrI, 0.590–2.354) but also between lorlatinib and chemotherapy (HR, 0.590; 95% CrI, 0.292–1.185), between lorlatinib and crizotinib (HR, 0.721; 95% CrI, 0.413–1.256), between lorlatinib and ceritinib (HR, 0.810; 95% CrI, 0.363–1.792), and between lorlatinib and brigatinib (HR, 0.736; 95% CrI, 0.305–1.759) (Figure 7).

The results of the comparison of OS between each pair of the six therapies, including comparison between lorlatinib and the other therapeutic agents are shown in Appendix A. The results of the ranking assessment based on SUCRA values showed that treatment with alectinib (SUCRA = 87.9%) had the highest for OS, followed by lorlatinib (SUCRA = 71.5%), ceritinib (SUCRA = 52.6%), brigatinib (SUCRA = 40.2%), crizotinib (SUCRA = 35.5%), and chemotherapy (SUCRA = 12.2%) (Appendix A).

Regarding OS, subgroup analysis by race (Asian or non-Asian), presence or absence of CNS metastasis, and PS (0–1) could not be performed because no corresponding data were reported.

### 3.7. Secondary Efficacy Endpoint: Proportions of Objective Response (ObRs)

All eight studies were included in the analysis of the ObRs. There were no significant differences in the ObRs between the lorlatinib-treated group and the alectinib-treated group (OR, 1.102; 95% CrI, 0.572–2.115). However, the ObR was significantly superior in the lorlatinib-treated group than in the chemotherapy-treated (OR, 10.49, 95% CrI, 5.583–19.61) and crizotinib-treated (OR, 2.292, 95% CrI, 1.391–3.768) groups. There were no significant differences in the ObRs between the lorlatinib-treated group and ceritinib-treated group (OR, 1.454; 95% CrI, 0.668–3.140) and lorlatinib-treated group and brigatinib-treated group (OR, 1.424; 95% CrI, 0.699–2.886). The results of the ObR comparison between each pair of the six therapies, including comparison between lorlatinib and other therapeutics, are shown in Appendix A.

The results of the ranking assessment using the SUCRA value showed that treatment with lorlatinib (SUCRA = 85.5%) ranked the highest, followed by alectinib (SUCRA = 78.7%), brigatinib (SUCRA = 57.5%), ceritinib (SUCRA = 56.3%) crizotinib (SUCRA = 22.0%), and chemotherapy (SUCRA = 0.0%) (Appendix A).

Regarding ObR, subgroup analysis by race (Asian or non-Asian), presence or absence of CNS metastasis, and PS (0–1) could not be performed because no corresponding data were reported.

### 3.8. Primary Safety Endpoint: Incidence of G3-AEs

For the primary safety endpoint, the incidence of G3-AEs, the data reported were not sufficient to make comparisons between the six treatment modalities included in this analysis (for the analyses of the G3-AEs, only five studies [ALEX, J-ALEX, ALESIA, ALTA-1L, and CROWN] were available for inclusion), and only comparisons among four treatments (crizotinib, alectinib, brigatinib, and lorlatinib) were possible.

The incidence of G3-AEs was more frequent in the lorlatinib group than in the alectinib group (RR, 1.918; 95% CrI, 1.486–2.475). In addition, the incidence of G3-AEs was more frequent in the lorlatinib group than in the crizotinib group (RR, 1.300; 95% CrI, 1.085–1.554). There were no significant differences in the incidence of G3-AEs between the lorlatinib-treated group and the brigatinib-treated group (RR, 1.181; 95% CrI, 0.900–1.546) (Figure 8).

The results of the comparison of G3-AEs-incidence between each pair of the four therapies (including comparison between lorlatinib and other therapeutics) are shown in Appendix A.

The results of the ranking assessment using SUCRA value for the incidence of G3-AEs showed that alectinib (SUCRA = 100.0%) was the safest, followed by crizotinib (SUCRA = 60.7%), brigatinib (SUCRA = 35.4%), and lorlatinib (SUCRA = 3.9%) (Appendix A).

### 3.9. Secondary Safety Endpoints: Incidence of AG-AEs, AG-SAEs, and G3-SAEs

For the secondary safety endpoints (AG-AEs, AG-SAEs, and G3-SAEs), the data reported were not sufficient to make comparisons among the six treatments included in this analysis (For the analysis of the AG-AEs, only five studies [ALEX, J-ALEX, ALESIA, ALTA-1L, and CROWN] were available for inclusion; for the analysis of AG-SAEs, four studies [ALEX, J-ALEX, ALESIA, and CROWN], and for G3-SAEs, two studies [ALEX and CROWN]). Therefore, for the comparison of AG-AEs, four treatment groups: crizotinib, alectinib, brigatinib, and lorlatinib were included; while for the comparison of AG-SAEs and G3-SAEs, only three treatment groups were included: crizotinib, alectinib, and lorlatinib.

There were no significant differences in the incidence of AG-AEs and G3-SAEs between the lorlatinib and alectinib groups (AG-AEs: RR, 1.018; 95% CrI, 0.985–1.051; G3-SAEs: RR, 1.255; 95% CrI, 0.737–2.146). The incidence of AG-SAEs was significantly higher in the lorlatinib-treated group than in the alectinib-treated group (RR, 1.614; 95% CrI, 1.042–2.503) (Appendix A).

The results of the comparison of the incidence of AG-AEs, AG-SAEs, and G3-AEs between each pair of groups treated with the therapies included in this analysis are also shown in Appendix A.

The results of the ranking assessment by using the SUCRA value revealed that, for the incidence of AG-AEs, brigatinib ranked as the most favorable (SUCRA = 94.4%), followed by alectinib (SUCRA = 58.0%), crizotinib (SUCRA = 34.9%), and lorlatinib (SUCRA = 12.7%); for the incidence of AG-SAEs, alectinib ranked as the most favorable (SUCRA = 97.8%), followed by crizotinib (SUCRA = 46.2%), and finally lorlatinib (SUCRA = 6.0%); and for the incidence of G3-SAEs, alectinib ranked as the most favorable (SUCRA = 68.4%), followed by crizotinib (SUCRA = 63.4%), and finally lorlatinib (SUCRA = 18.3%). The results of the ranking assessment for each of these secondary endpoints (AG-AEs, AG-SAEs, and G3-SAEs) are summarized in Appendix A.

Alectinib was the most favorable in terms of G3-AEs, AG-SAEs, and G3-SAEs and ranked second for AG-AEs, while lorlatinib ranked the lowest for AG-AEs, G3-AEs, AG-SAEs, and G3-SAEs.

### 3.10. Secondary Safety Endpoints: Incidence of AG-Nausea, G3-Nausea, AG-Diarrhea, and G3-Diarrhea

All eight studies were included in the analysis of AG-nausea and AG-diarrhea, but only seven studies [PROFILE1014, ASCEND-4, ALEX, J-ALEX, ALESIA, ALTA-1L, and CROWN] were available for the analysis of G3-nausea. For the analysis of G3-diarrhea, only six studies [PROFILE1014, ASCEND-4, ALEX, J-ALEX, ALTA-1L, and CROWN] could be included. There were no significant differences in the incidence of AG-nausea, G3-nausea, and G3-diarrhea between the lorlatinib and alectinib groups (AG-nausea: RR, 1.284; 95% CrI, 0.764–2.153; G3-nausea: RR, 1.487; 95% CrI, 0.097–22.690; G3-diarrhea: RR, 11.620; 95% CrI, 0.482–275.70). The incidence of AG-diarrhea was significantly higher in the lorlatinib-treated group than in the alectinib-treated group (RR, 1.869; 95% CrI, 1.167–2.988) (Appendix A).

The results of the comparison of the incidence of AG-nausea, G3-nausea, AG-diarrhea, and G3-diarrhea between each pair of groups treated with the therapies included in this analysis are also shown in Appendix A.

The results of the ranking assessment for each of these secondary endpoints (AG-AEs, AG-SAEs, and G3-SAEs) are summarized in Appendix A. Alectinib ranked most favorable in terms of AG-nausea, G3-nausea, and G3-diarrhea and second most favorable in terms of AG-diarrhea.

### 3.11. Secondary Safety Endpoints: Incidence of AG-AST, G3-AST, AG-ALT, and G3-ALT

For the secondary safety endpoints (AG-AST, G3-AST, AG-ALT, and G3-ALT), the data reported were not sufficient to make comparisons among the six treatments included in this analysis (only four studies [ALEX, J-ALEX, ALTA-1L, and CROWN] could be included in the analysis of AG-AST and G3-AST, and five studies [ALEX, J-ALEX, ALESIA, ALTA-1L, and CROWN] in the analysis of AG-ALT and G3-ALT). Therefore, for the comparison of AG-AST, G3-AST, AG-ALT, and G3-ALT, four treatment groups: crizotinib, alectinib, brigatinib, and lorlatinib were included.

There were no significant differences in the incidence of AG-AST, G3-AST, AG-ALT, and G3-ALT between the lorlatinib and alectinib groups (AG-AST: RR, 1.089; 95% CrI, 0.591–2.006; G3-AST: RR, 1.269; 95% CrI, 0.254–6.300; AG-AST: RR, 0.869; 95% CrI, 0.538–1.399; G3-AST: RR, 2.447; 95% CrI, 0.584–10.18) (Appendix A). The results of the comparison of the incidence of AG-AST, G3-AST, AG-ALT, and G3-ALT between each pair of groups treated with the therapies included in this analysis are also shown in Appendix A. The results of the ranking assessment by using the SUCRA value were shown in Appendix A. Crizotinib ranked the lowest in terms of AG-AST, G3-AST, AG-ALT, and G3-ALT.

### 3.12. Secondary Safety Endpoinst: Incidence of AG-Pneumonitis and G3-Pneumonitis

For the secondary safety endpoints (AG-pneumonitis and G3-pneumonitis), the data reported were not sufficient to make comparisons among the six treatments included in this analysis (only two studies [ALEX and CROWN] could be included in the analysis of AG-pneumonia and G3-pneumonia). Therefore, for the comparison of AG-pneumonitis and G3-pneumonitis, only three treatment groups (crizotinib, alectinib, and lorlatinib) were included.

There were no significant differences in the incidence of AG-pneumonitis and G3-pneumonitis between the lorlatinib and alectinib groups (AG-pneumonitis: RR, 1.881; 95% CrI, 0.145–25.02; G3-pneumonitis: RR, 2.250; 95% CrI, 0.029–177.2) (Appendix A). The results of the comparison of AG-pneumonitis and G3-pneumonitis incidence between each pair of groups treated with the therapies included in this analysis are also shown in Appendix A. The results of the ranking assessment by using the SUCRA value were shown in Appendix A. In terms of both AG-pneumonitis and G3-pneumonia, alectinib ranked the most favorable, followed by lorlatinib and crizotinib.

### 3.13. Bias Assessment

The Cochrane-recommended RoB2 was used to assess the quality of the included studies [48]. All eight studies included in the current systematic review and NMA were judged to have “some concerns” in the overall assessment. This was because all of them were open-label studies and were judged to have some concerns with regard to bias due to deviation from intended intervention and bias in measurement of the outcome. Additionally, PROFILE1029 [61] was judged to have some concerns in the domain of bias arising from the randomization process owing to inadequate descriptions of the details of randomization. There were no studies with domains that were considered high risk (Appendix A).

### 3.14. Sensitivity Analysis

Of the eight studies included in our systematic review and NMA, the two studies partially included a group of patients who had received previous chemotherapy. To address this heterogeneity, we performed a sensitivity analysis by excluding the patient population with prior chemotherapy from the two studies (ALTA-1L and J-ALEX). The results showed that there was no significant difference between lorlatinib and alectinib in terms of PFS, and the results of the significance assessment comparing each pair of the six treatment arms were maintained (Appendix A). Furthermore, the same results were obtained for the ranking of the six treatment groups (Appendix A). Considering these results, we concluded that the inclusion or exclusion of patients who had been previously treated with chemotherapy would not affect our final conclusion.

### 3.15. Assessment of Between-Study Heterogeneity

Both PROFILE1014 [60] and PROFILE1029 [61] compared crizotinib and chemotherapy. Therefore, the between-study heterogeneity in PFS was evaluated in these two trials. The results showed that the *I*^2^ had a value of 0.0% (*p* = 0.582), indicating no statistically significant difference in the between-study heterogeneity (Appendix A).

Furthermore, ALEX [22], J-ALEX [23], and ALESIA [62] compared alectinib and crizotinib. Therefore, the between-study heterogeneity in PFS was also evaluated in these three trials. The results showed that *I*^2^ had a value of 65.0% (*p* = 0.057), indicating the presence of significant between-study heterogeneity (Appendix A).

## 4. Discussion

In this study, we performed an NMA that included six treatment arms: lorlatinib, brigatinib, alectinib, ceritinib, crizotinib, and chemotherapy with the main objective of comparing the efficacy and safety of lorlatinib and alectinib in ALK-p ALK-inhibitor-naïve advanced NSCLC patients.

The results showed that there were no significant differences in the efficacy of lorlatinib and alectinib with regard to PFS and OS. The racial subgroup analyses showed that there were no significant differences in PFS between lorlatinib and alectinib in Asians, whereas in non-Asians, PFS was significantly more favorable with lorlatinib than with alectinib. The ranking assessment among the six treatment arms lorlatinib, brigatinib, alectinib, ceritinib, crizotinib, and platinum-based chemotherapy revealed that lorlatinib ranked the highest for efficacy in PFS, whereas alectinib ranked the highest for efficacy in OS. As for the racial ranking assessment for PFS, in non-Asian patients, lorlatinib ranked the highest for efficacy, whereas in Asian patients, alectinib ranked the highest in efficacy. With regard to safety outcomes, the incidence of G3-AEs was significantly higher with lorlatinib than with alectinib. The results of the ranking assessment for G3-AEs among the four treatment arms of lorlatinib, brigatinib, alectinib, and crizotinib revealed that alectinib was the safest and that lorlatinib ranked the lowest in this regard.

The NCCN guidelines and the Japanese guidelines for the treatment of lung cancer recommend alectinib as first-line therapy for advanced NSCLC without ALK-p ALK inhibitors, whereas lorlatinib is not approved as a first-line therapy. However, in a previous large RCT of ALK-p advanced NSCLC, lorlatinib was reported to have a superior efficacy compared to crizotinib as a first-line therapy [33]. However, although lorlatinib is expected to be a new first-line treatment for ALK-p advanced NSCLC, there have been no RCTs comparing its efficacy with that of alectinib, the current first-line treatment for ALK-p advanced NSCLC. To the best of our knowledge, this is the first report that compared the efficacy and safety of lorlatinib and alectinib using NMA with Bayesian statistical methods, not only in the overall group of participants but also in the subgroups of Asian and non-Asian patients. The results showed that there were no significant differences in PFS and OS between lorlatinib and alectinib in the analysis of the overall group of participants. However, although there were no significant differences in PFS between lorlatinib and alectinib in the subgroup of Asian patients, in non-Asian patients, the PFS in the lorlatinib group was significantly higher than that in the alectinib group. More notably, in terms of PFS efficacy, lorlatinib ranked as the most effective treatment not only in the analysis of the entire study population, but also in the subgroup analysis of patients with and without CNS metastases, which was a secondary endpoint in this analysis. Furthermore, lorlatinib ranked highest in efficacy for ObRs, which was also a secondary endpoint. These results suggest that lorlatinib may be a promising new therapeutic option for the first-line treatment of ALK-naïve ALK-p advanced NSCLC.

Brigatinib, like lorlatinib, has been shown to be effective against resistant mutations of ALK inhibitors, and, in addition, has been reported to be effective in patients with CNS metastases [65,66,67]. A previous large RCT showed that brigatinib was more effective for ALK-p ALK-inhibitor-naïve advanced NSCLC than crizotinib [64]. Brigatinib, like alectinib, is a promising first-line treatment option for ALK-p advanced NSCLC [68]. Therefore, we believe that the comparison between lorlatinib and brigatinib provides important information for improving treatment strategies for ALK-p ALK-inhibitor-naïve advanced NSCLC. Our NMA showed that in all participants and in the subgroup without CNS metastasis, although there were no significant differences between lorlatinib and alectinib, lorlatinib administration resulted in significantly superior PFS than that of chemotherapy, crizotinib, ceritinib, and brigatinib. Even in the subgroup with CNS metastases, lorlatinib ranked the highest in terms of efficacy for PFS. These results suggest that lorlatinib has a relatively favorable efficacy profile compared to other existing ALK inhibitors, and support the possibility that lorlatinib may be a potential new first-line treatment option for ALK-p advanced NSCLC.

These results also suggest that the efficacy of lorlatinib may be explained by analyzing the molecular biology of the disease and drugs. It has long been pointed out that it is challenging for conventional ALK-inhibitors to maintain sufficient efficacy against brain metastases owing to their low permeability of the blood–brain barrier [17,69]. Lorlatinib had been designed to penetrate the blood–brain barrier [28,29]; therefore, it was expected to be more effective than conventional ALK-inhibitors for the treatment of brain metastases. Moreover, lorlatinib has also been shown to reduce the size of brain metastases in in vivo mouse models [28,31]. Furthermore, phase I and II studies have shown that the ObRs of lorlatinib in patients with brain metastases were favorable [70,71]. More significantly, long-term treatment with alectinib has been reported to induce resistant mutations, such as G1202R and I1171N [19,20,25]. In other words, there is concern regarding the efficacy of alectinib, which is seen to gradually decrease with long-term administration. The molecular structure of lorlatinib has been designed to maintain its efficacy against tumor cells with such resistant mutations [28,29,72] (Figure 9). These properties of lorlatinib may strongly support its efficacy.

Nonetheless, our results do not necessarily indicate that lorlatinib is the best first-line treatment in all ALK-inhibitor naïve advanced NSCLC cases. Our analysis revealed that although there was no significant difference in OS between lorlatinib and crizotinib or between lorlatinib and chemotherapy, alectinib showed a significantly more favorable OS than crizotinib or than chemotherapy. In addition, the ranking assessment showed that alectinib ranked the highest for PFS in the Asian subgroup and for OS in the overall group of participants, respectively. Furthermore, the incidence of G3-AEs was significantly higher in the lorlatinib group than in the alectinib group. These results support the continued recommendation of alectinib as one of the standard first-line therapeutic options for ALK-naïve ALK-positive advanced NSCLC even if lorlatinib becomes available for ALK-naïve patients. We consider that the appropriateness of lorlatinib and alectinib in ALK-p ALK-inhibitor–naïve advanced NSCLC needs to be evaluated further in the future. Further clinical studies are needed to identify patient populations and predictors of efficacy in which lorlatinib or alectinib is expected to be effective for ALK-inhibitor naïve ALK-p advanced NSCLC.

The results of our subgroup analyses by ethnicity showed that lorlatinib had a significantly superior PFS than alectinib in the non-Asian subgroup, while there was no significant difference in PFS between the two drugs in the Asian subgroup. A novel finding of our study is that lorlatinib may be superior to alectinib in the subgroup of non-Asian patients with ALK-p ALK-naïve advanced NSCLC.

It is extremely difficult to explain the racial differences in the efficacy of lorlatinib on the basis of molecular biology. Previous reports have shown that the efficacy of lorlatinib in previously treated ALK patients was influenced by the EML4-ALK variant. Variant 3 is generally reported to be less effective with ALK inhibitors than variant 1 and is associated with poor prognosis. However, the median PFS of upon treatment with lorlatinib was 3.3 months for patients with variant 1 as against 11.0 months for patients with variant 3, and a significant reduction of 69% was observed in the risk of disease progression or death in the variant 3 group compared with that in the variant 1 group [73]. This means that lorlatinib showed higher clinical efficacy in patients with variant 3 rather than in patients with variant 1. Furthermore, the proportion of patients with the G1202R mutation was significantly higher in the variant 3 group (32%) than in the variant 1 group (0%). In other words, the incidence of G1202R, for which lorlatinib has a higher affinity and sensitivity compared with other ALK inhibitors, was higher in the variant 3 group; in a population with a higher percentage of patients in the variant 3 group, lorlatinib is expected to have higher efficacy than other ALK inhibitors [73]. However, whether the ratio of EML4-ALK variants differs between Asians and non-Asians has not been fully assessed. Further work is needed to validate these findings.

In addition, subset analysis of OS and safety by race could not be performed in the current analysis because the data required for the analysis were not reported. Therefore, we were unable to assess whether the efficacy of lorlatinib on PFS in non-Asians was maintained for OS as well and whether the frequency of adverse events differed by race in this analysis. More research assessing racial differences in lorlatinib efficacy for OS and safety profiles, in addition to PFS, would be helpful for developing race-specific treatment strategies.

With regard to the safety profile, although there was no significant difference in the incidence of pneumonitis, increased AST or ALT, or nausea between lorlatinib and alectinib, regardless of the severity (any grade or Grade 3 or more), the incidence of not only G3-AEs but also AG-SAEs and AG-diarrhea was significantly more frequent in lorlatinib than in alectinib. Furthermore, in a safety ranking of the four treatment groups (crizotinib, alectinib, brigatinib, and lorlatinib), alectinib was the safest and lorlatinib was the least safe for G3-AEs, AG-SAEs, and G3-SAEs (regarding AG-AEs, alectinib was the second safest and lorlatinib was the least safe). The results of these analyses for safety profiles have provided valuable information for the development of therapeutic strategies for ALK inhibitor naïve ALK-p advanced NSCLC, suggesting that particularly high and adequate safety considerations may be required when lorlatinib is administered.

Several previous meta-analyses and NMAs have compared the efficacy and safety of ALK inhibitors in ALK-p ALK-inhibitor–naïve advanced NSCLC [65,74,75,76,77,78,79,80]. However, there have been no previous reports that have compared PFS as well as OS in evaluating the efficacy profile not only by race but also by the presence of brain metastases, PS, and prior therapies, or evaluating the detailed safety profile. Here, for the first time, we demonstrated that PFS and OS did not differ significantly under lorlatinib versus alectinib treatment, and that, when looking within patient subgroups, lorlatinib improved PFS more than alectinib in non-Asians, but not in Asians. The incidence of G3-AEs was higher in the lorlatinib treated group than in the alectinib treated group. Alectinib ranked most favorably for OS and G3-AEs, whereas lorlatinib ranked the lowest for G3-AEs. The results we present here may be valuable for clinical oncologists in their development and implementation of future treatment strategies for ALK-inhibitor naïve ALP-p advanced NSCLC. Since an NMA consists of direct and indirect comparisons, it is difficult to draw definitive final conclusions from these results; however, they support the possibility that lorlatinib may be one of emerging first-line treatment options for ALK-p and ALK inhibitor-naïve advanced NSCLC. On the other hand, we consider that alectinib also will continue to be one of the leading first-line treatment options for ALK-naïve ALK-p advanced NSCLC. Further validation is required to provide guidance on how to differentiate between lorlatinib and alectinib for advanced ALK-p NSCLC and at what stage of the treatment sequence (first-line, second-line, etc.) lorlatinib will be the most beneficial to patients. A better understanding of the molecular biology and clinical aspects of cancers with ALK rearrangement is needed to determine the best treatment strategy [81]. Additional studies, including analysis of accumulated data from future large-scale, direct comparison RCTs, will therefore be required.

There were several limitations of this NMA that should be recognized. First, this was a systematic review, and the NMA compared the efficacy and safety of lorlatinib and alectinib and analyzed the results by race. However, the race-specific analysis of OS and G3-AEs was not possible due to the lack of reported data. Further verification is needed to confirm whether the results of the present analysis of PFS by race will maintain the same trend (lorlatinib efficacy in non-Asian patients) for OS and whether G3-AEs differ by race. Next, this analysis included patients with a PS of 0 to 2. Seven studies used the ECOG-PS as a criterion for evaluating PS, and one reference (ASCEND-4 [63]) used the WHO-PS. Furthermore, the percentage of patients with a PS of 2 was different in each of the studies included in this analysis. We performed a predefined subgroup analysis that included only patients with ECOG-PS between 0 and 1, excluded the group of patients with a PS of 2, and the group of patients included in ASCEND-4 [63]. Although, judging from the results of this subgroup analysis, while we considered that this heterogeneity was appropriately addressed and that it did not affect our final conclusions, we cannot completely exclude the possibility that it had a non-negligible effect. Third, the patient group studied in this analysis included patients who received systemic anticancer chemotherapy and those who did not. According to the results of the sensitivity analysis performed to address this heterogeneity, the inclusion and exclusion of patients with prior systemic anticancer therapy was not expected to affect the final conclusions; however, the impact of this heterogeneity cannot be ignored. Finally, we found heterogeneity, although not statistically significant, between the three studies comparing alectinib and crizotinib (ALEX [22], J-ALEX [23], and ALESIA [62]). Although the Bayes model, which assumes potential heterogeneity among the included studies, was used in this NMA, we cannot completely exclude the possibility that it influenced the final conclusion.

## 5. Conclusions

In summary, this systematic review and NMA compared the efficacy and safety of lorlatinib and alectinib in ALK-p and ALK inhibitor-naïve NSCLC patients according to the best practice guidelines. There was no significant difference in PFS and OS between the lorlatinib and alectinib groups. In the subgroup analyses, although there was no significant difference in PFS between lorlatinib and alectinib in the subgroup of Asian patients, in the non-Asian patient subgroup, PFS was significantly superior with lorlatinib than with alectinib. The incidence of G3-AEs was also significantly higher in the lorlatinib group than in the alectinib group. In a comparison of the six treatment groups (lorlatinib, brigatinib, alectinib, ceritinib, crizotinib, and chemotherapy), lorlatinib ranked the highest for PFS, followed by alectinib, brigatinib, crizotinib, ceritinib, and chemotherapy, whereas in terms of OS, alectinib ranked the highest, followed by lorlatinib, ceritinib, brigatinib, crizotinib, and chemotherapy. In terms of G3-AEs incidence, alectinib was the safest, followed by crizotinib, brigatinib, and lorlatinib. The current study was an NMA that included direct and indirect comparisons, and the results of this validation need to be confirmed in large-scale head-to-head RCTs using direct comparisons. Further studies are required to provide information on patient background and predictors of efficacy for the clinical question on the use of lorlatinib or alectinib for ALK-p ALK-inhibitor-naïve advanced NSCLC.

## Figures and Tables

**Figure 1 cancers-13-03704-f001:**
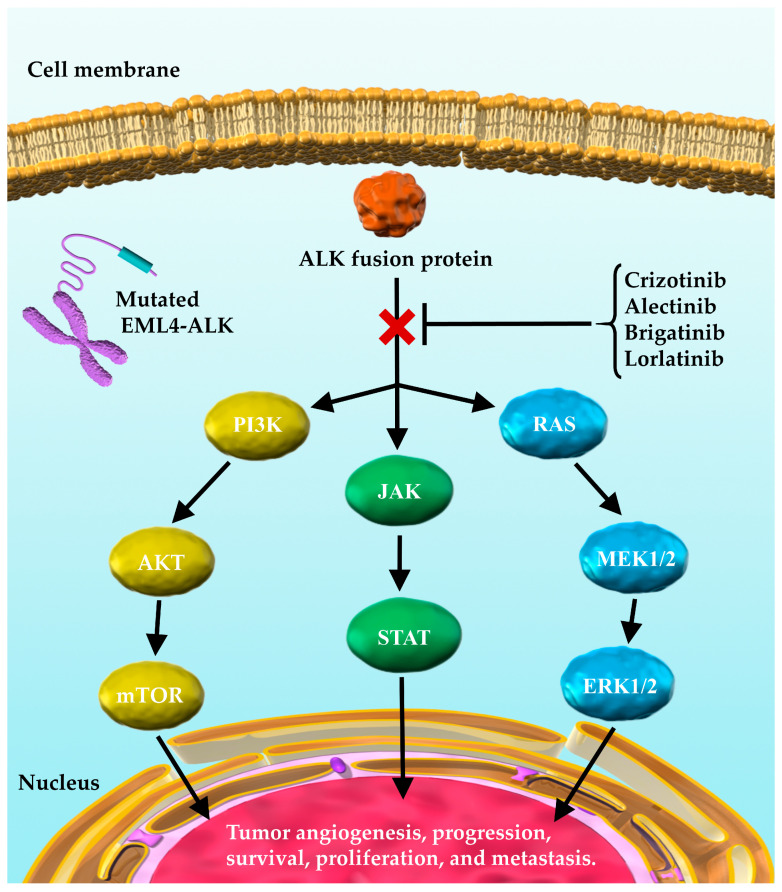
ALK fusion proteins and their downstream signaling pathways involved in tumor progression. EML4-ALK activates PI3K-AKT, RAS, and JAK/STAT signaling cascades through the ALK fusion protein. This in turn affects tumor progression, survival, and growth. ALK inhibitors, such as crizotinib, alectinib, brigatinib, and lorlatinib exert their antitumor effects by acting on mutated ALK and inhibiting the effects of ALK fusion proteins on downstream signaling. EML4-ALK, echinoderm microtubule-associated protein-like 4-anaplastic lymphoma kinase; ALK, anaplastic lymphoma kinase; PI3K, phosphatidylinositol-3 kinase; mTOR, mammalian target of rapamycin; RAS, reticular activating system; MEK, mitogen-activated extracellular signal regulated kinase; ERK, extracellular signal-regulated kinase (ERK); JAK, Janus kinase; STAT, signal transducer and activator of transcription.

**Figure 2 cancers-13-03704-f002:**
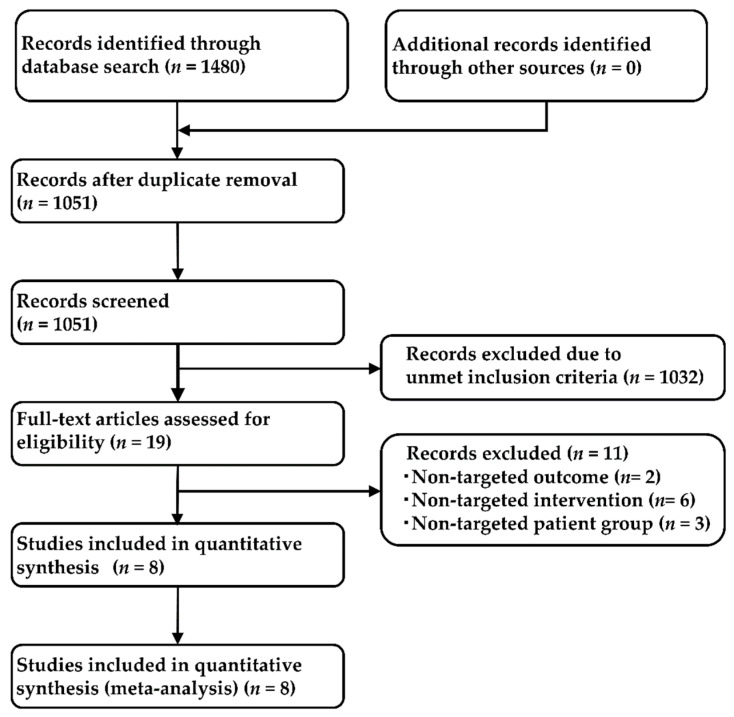
Flow chart depicting the process used for study selection.

**Figure 3 cancers-13-03704-f003:**
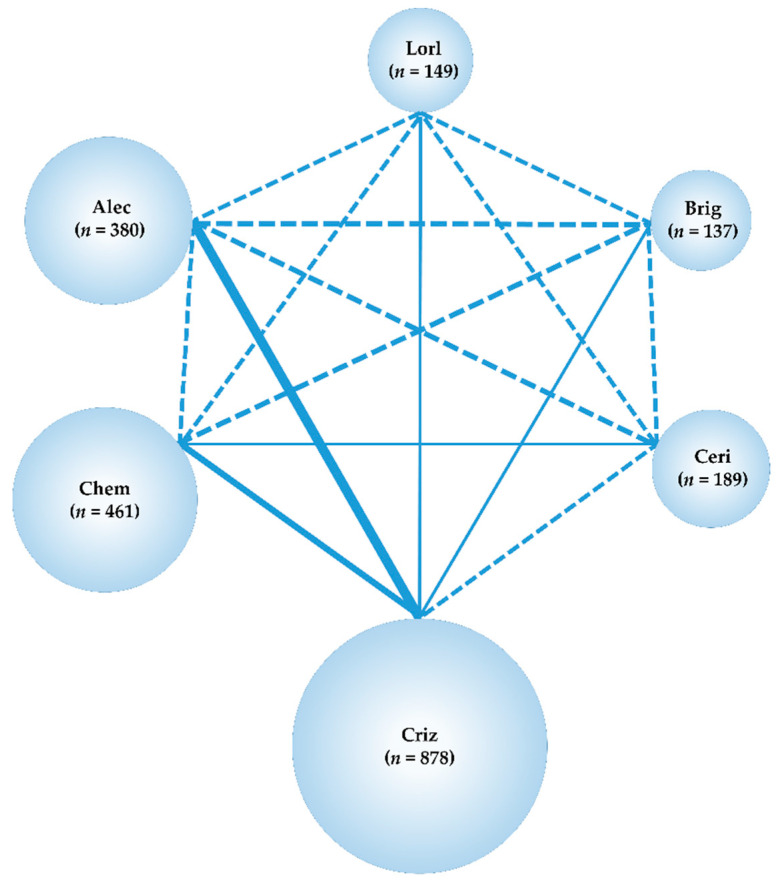
Network map. Network map of the six treatment arms i.e., lorlatinib, brigatinib, alectinib, ceritinib, crizotinib, and chemotherapy. The randomized controlled trials (RCTs) included in the network meta-analysis (NMA) are shown as solid lines, and the width of the solid line corresponds to the number of included trials. Dashed lines indicate that there are no head-to-head RCTs and that treatment comparisons will be attempted. *n* is the number of patients included in each arm. Lorl, lorlatinib; Brig, brigatinib; Alec, alectinib; Ceri, ceritinib; Criz, crizotinib; Chem, chemotherapy RCT, randomized controlled trial; NMA, network meta-analysis.

**Figure 4 cancers-13-03704-f004:**
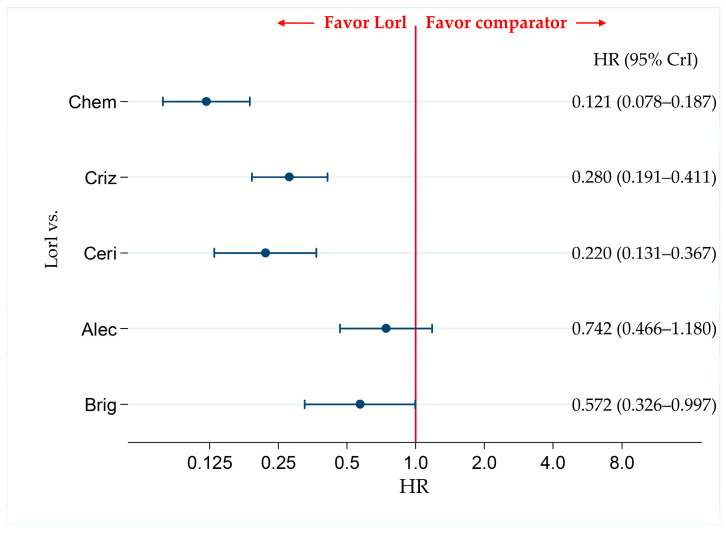
Comparative efficacy of lorlatinib and existing therapeutics in terms of PFS in patients with ALK-p ALK-inhibitor-naïve NSCLC. All eight studies were included for analyzing PFS in the overall participant group. A comparison of PFS in ALK-p ALK-inhibitor-naïve advanced NSCLC patients treated with lorlatinib and each of the other five therapeutic agents including chemotherapy, crizotinib, ceritinib, alectinib, and brigatinib as comparator is presented. Comparisons are expressed as lorlatinib versus each of the comparators. Data are expressed as hazard ratios (HRs) and 95% credible intervals (CrIs). PFS, progression free survival; ALK, anaplastic lymphoma kinase; ALK-p, anaplastic lymphoma kinase-positive; NSCLC, non-small cell lung cancer; Lorl, lorlatinib; Brig, brigatinib; Alec, alectinib; Ceri, ceritinib; Criz, crizotinib; Chem, chemotherapy.

**Figure 5 cancers-13-03704-f005:**
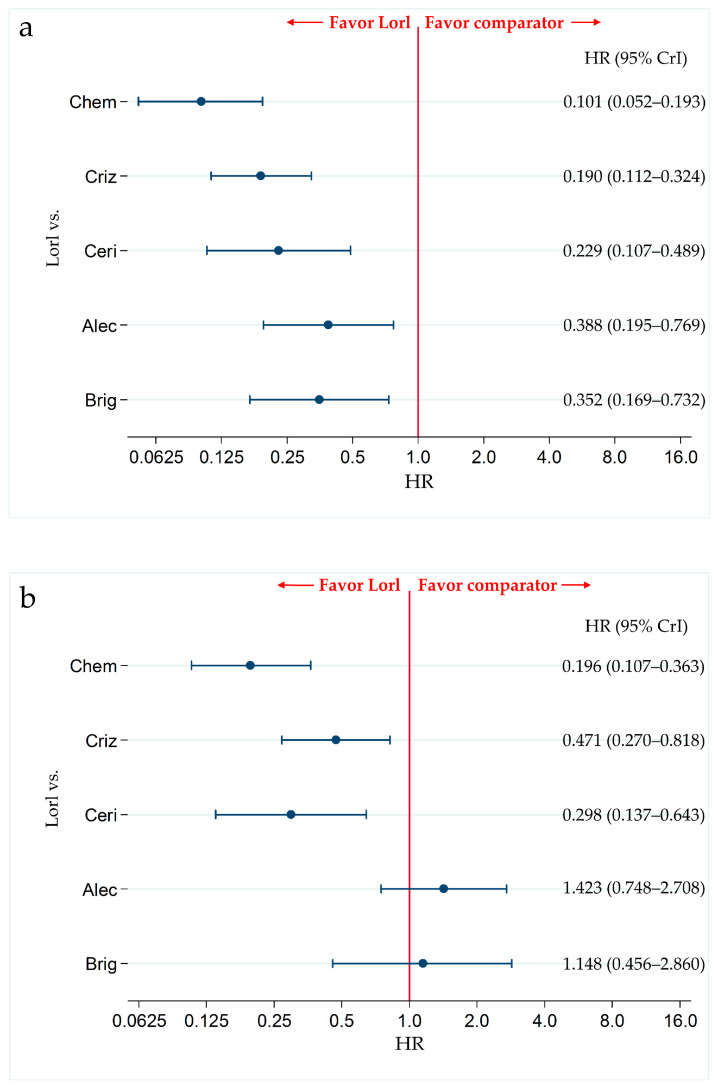
Comparative efficacy of lorlatinib and existing therapeutics in terms of PFS with in non-Asian and Asian patients with ALK-p ALK-inhibitor-naïve advanced NSCLC. A comparison between lorlatinib and each of the other five therapeutic agents, including chemotherapy, crizotinib, ceritinib, alectinib, and brigatinib as comparator in terms of PFS in patients with ALK-p ALK-inhibitor-naïve advanced NSCLC is presented. Comparisons are expressed as lorlatinib versus each of the comparator agents. Data are expressed as hazard ratios (HRs) and 95% credible intervals (CrIs); (**a**) Comparison in the non-Asian subgroup; (**b**) Comparison in the Asian subgroup. PFS, progression free survival; ALK, Anaplastic Lymphoma Kinase; ALK-p, Anaplastic Lymphoma Kinase-positive; NSCLC, non-small cell lung cancer; Lorl, lorlatinib; Brig, brigatinib; Alec, alectinib; Ceri, ceritinib; Criz, crizotinib; Chem, chemotherapy.

**Figure 6 cancers-13-03704-f006:**
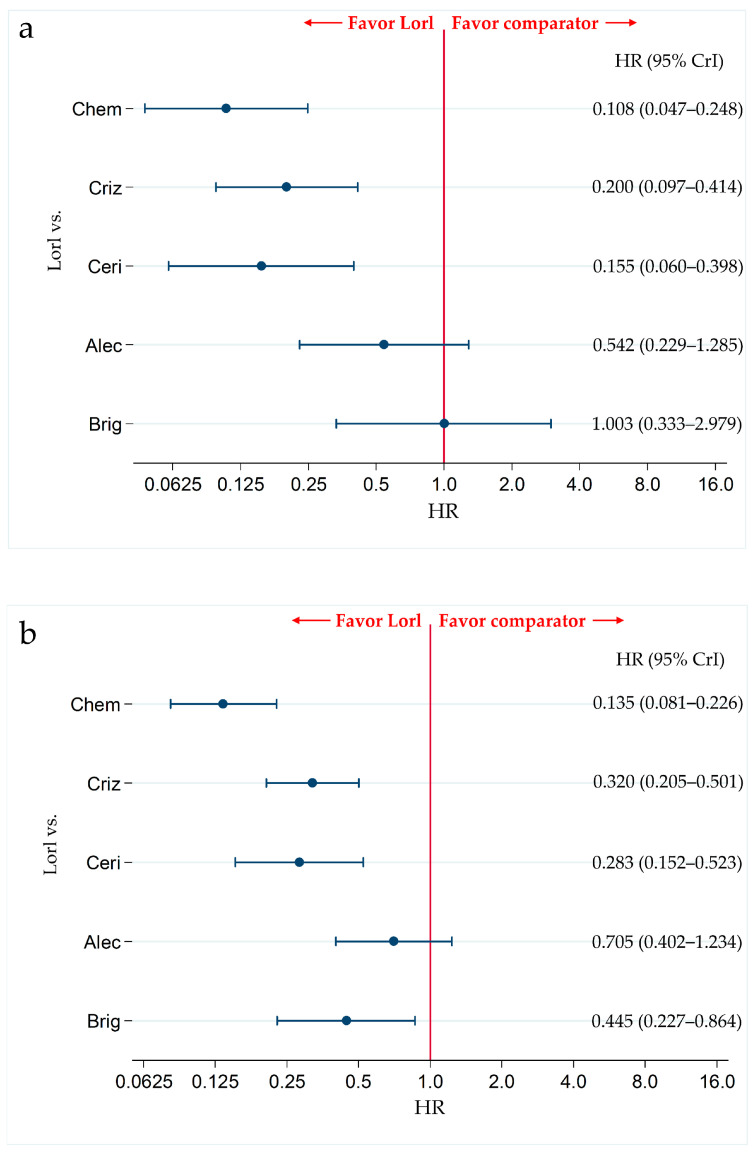
Comparative efficacy of lorlatinib and existing therapeutics in terms of PFS in ALK-p ALK-inhibitor-naïve advanced NSCLC patients with and without CNS metastasis. A comparison between the PFS of patients with ALK-p ALK-inhibitor-naïve advanced NSCLC who were treated with lorlatinib and each of the other five therapeutic agents, chemotherapy, crizotinib, ceritinib, alectinib, and brigatinib as comparator is presented. Comparisons are expressed as lorlatinib versus each of the comparator agents. Data are expressed as hazard ratios (HRs) and 95% credible intervals (CrIs); (**a**) Comparison in the subgroup of patients with CNS metastases; (**b**) Comparison in the subgroup of patients without CNS metastases. PFS, progression-free survival; ALK, Anaplastic Lymphoma Kinase; ALK-p, Anaplastic Lymphoma Kinase-positive; NSCLC, non-small cell lung cancer; Lorl, lorlatinib; Brig, brigatinib; Alec, alectinib; Ceri, ceritinib; Criz, crizotinib; Chem, chemotherapy.

**Figure 7 cancers-13-03704-f007:**
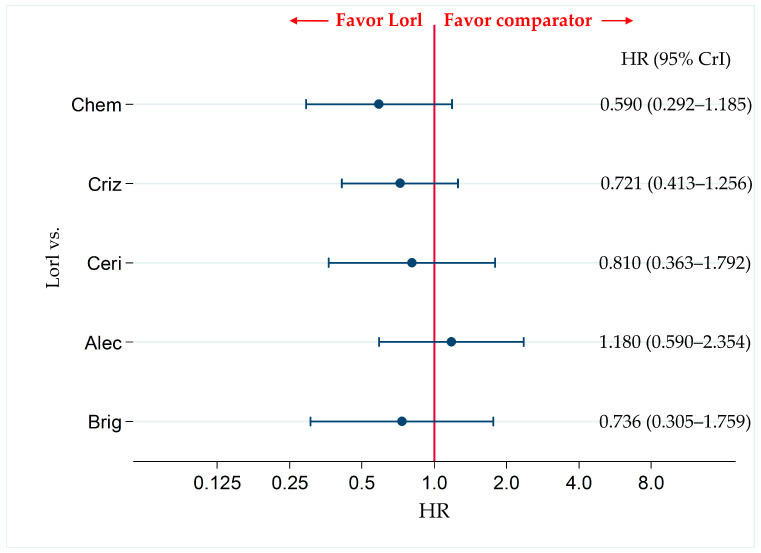
Comparative efficacy of lorlatinib and the existing therapeutic agents in terms of OS in patients with ALK-p ALK-inhibitor-naïve NSCLC. A comparison of OS in ALK-p ALK-inhibitor-naïve advanced NSCLC patients treated with lorlatinib and each of the other five therapeutic agents including chemotherapy, crizotinib, ceritinib, alectinib, and brigatinib as comparator is presented. Comparisons are expressed as lorlatinib versus each of the comparator agents. Data are expressed as hazard ratios (HRs) and 95% credible intervals (CrIs). OS, overall survival; ALK, anaplastic lymphoma kinase; ALK-p, anaplastic lymphoma kinase-positive; NSCLC, non-small cell lung cancer; Lorl, lorlatinib; Brig, brigatinib; Alec, alectinib; Ceri, ceritinib; Criz, crizotinib; Chem, chemotherapy.

**Figure 8 cancers-13-03704-f008:**
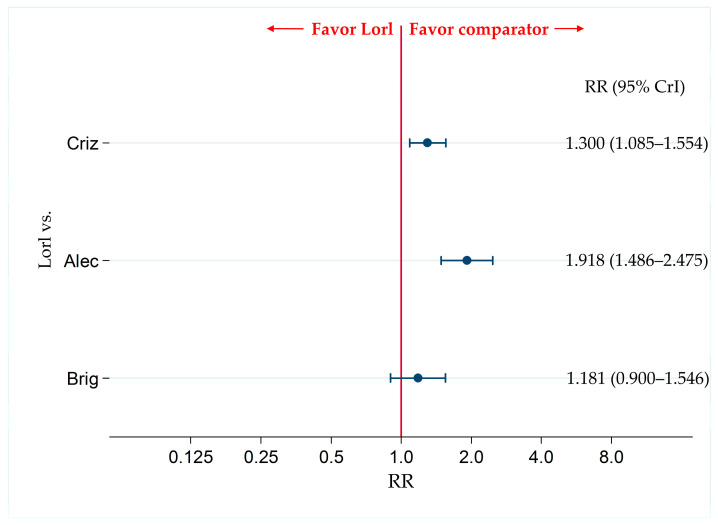
Comparative safety of G3-AEs with lorlatinib and the existing therapeutic agents for ALK-p ALK-inhibitor-naïve advanced NSCLC. Comparison of safety by assessing G3-AEs for ALK-p ALK-inhibitor-naïve advanced NSCLC between lorlatinib and each of the three other therapeutic agents including crizotinib, alectinib, and brigatinib as comparator are presented. Comparisons are expressed as lorlatinib versus each of the comparator agents. Data are expressed as risk ratios (RRs) and 95% credible intervals (CrIs) G3-AEs, grade 3 or higher adverse events; ALK, anaplastic lymphoma kinase; ALK-p, anaplastic lymphoma kinase-positive; NSCLC, non-small cell lung cancer; Lorl, lorlatinib; Brig, brigatinib; Alec, alectinib; Criz, crizotinib.

**Figure 9 cancers-13-03704-f009:**
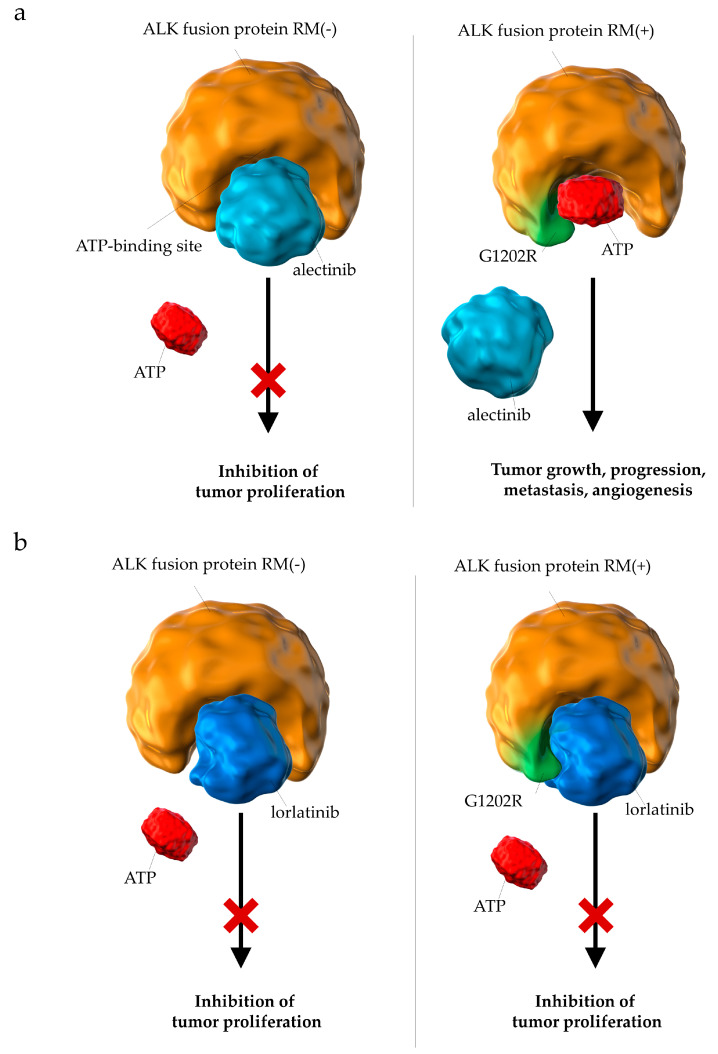
Differences in the pharmacological effects between (**a**) alectinib and (**b**) lorlatinib on ALK fusion proteins with resistant mutations. (**a**) Resistant mutations (e.g., G1202R) prevent the binding of alectinib to the ATP-binding domain of ALK fusion proteins. (**b**) Lorlatinib binds to the ATP-binding pocket of ALK fusion proteins with resistant mutations and successfully suppresses downstream signals involved in tumor growth; ALK, anaplastic lymphoma kinase; RM, resistant mutation; ATP, adenosine triphosphate.

## Data Availability

The authors confirm that the data sets analyzed in the present study are available from the corresponding author upon reasonable request.

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
