# Peer review of "Comparative Efficacy and Safety of Lorlatinib and Alectinib for ALK-Rearrangement Positive Advanced Non-Small Cell Lung Cancer in Asian and Non-Asian Patients: A Systematic Review and Network Meta-Analysis"

_cancers, 2021, doi:10.3390/cancers13153704_

Round 1
Reviewer 1 Report
In their manuscript Koichi and colleagues describe the results of a systematic review and network metanalysis about six different treatments in ALK positive NSCLC. The topic is of potential interest for the readers and the overall quality of the manuscript is good. I have no specific comment.
Author Response
Dear Reviewer 1
Thank you for giving us the opportunity to resubmit to Cancers a revised draft of our manuscript titled “Comparative Efficacy and Safety of Lorlatinib and Alectinib for ALK-Rearrangement Positive Advanced Non-Small Cell Lung Cancer in Asian and Non-Asian Patients: A Systematic Review and Network Meta-Analysis” (Manuscript ID: cancers-1294387). We sincerely appreciate the time and effort that you and the reviewers have dedicated to providing your valuable feedback on our manuscript. We are grateful to the reviewers for their insightful comments. We have made some minor revisions to some parts of our manuscript according to the reviewers' comments. We have highlighted in yellow the changes within the manuscript.
The revised parts are as follows.
Lines 261, 262, and 263: 
“I2” was changed to “I2.”
Line 682:
"resistant @mutations" has been changed to "resistant mutations."
No revisions were made to the Figures, Tables, or Supplementary information.
Here is a response to the comments of the Reviewer 1.
Comment1: In their manuscript Koichi and colleagues describe the results of a systematic review and network meta-analysis about six different treatments in ALK positive NSCLC. The topic is of potential interest for the readers and the overall quality of the manuscript is good. I have no specific comment.
Response1: We wish to express our strong appreciation to the reviewers for their insightful comments on our manuscript.
We are confident that our revised manuscript will be suitable for publication in Cancers and look forward to receiving your editorial decision.
Thank you for your consideration.
Sincerely,
Koichi Ando
Department of Medicine, Division of Respiratory Medicine and Allergology, Showa University School of Medicine
1-5-8 Hatanodai, Shinagawa-ku, Tokyo, 142-8666, Japan
Tel: +81-3-3784-8532
Fax: +81-3-3784-8742
Email: koichi-a@med.showa-u.ac.jp
Reviewer 2 Report
I reviewed the Systematic Review and Network Meta-Analysis entitled “Comparative Efficacy and Safety of Lorlatinib and Alectinib for ALK-Rearrangement Positive Advanced Non-Small Cell Lung Cancer in Asian and Non-Asian Patients”, which is under consideration for publications in Cancers.
Overall, it is a comprehensive review where the author has included data from 1480 studies related to treatment of non-small cell lung cancer. By these analysis, they have compared Lorlatinib to five other treatment options for non-small cell lung cancer, with a special focus on Alectinib.
The review is well balanced and includes treatment outcomes and side effects to the different treatments. The authors has also included different tumor stages and metastasis to the brain, where Lorlatinib shows advantages.
The authors discuss the limitation in a systematic review and limitation in available information, which can have impact on the conclusion. Overall, I find the study ready for publication.
Author Response
Dear Reviewer 2
Thank you for giving us the opportunity to resubmit to Cancers a revised draft of our manuscript titled “Comparative Efficacy and Safety of Lorlatinib and Alectinib for ALK-Rearrangement Positive Advanced Non-Small Cell Lung Cancer in Asian and Non-Asian Patients: A Systematic Review and Network Meta-Analysis” (Manuscript ID: cancers-1294387). We sincerely appreciate the time and effort that you and the reviewers have dedicated to providing your valuable feedback on our manuscript. We are grateful to the reviewers for their insightful comments. We have made some minor revisions to our manuscript according to the reviewers' comments. We have highlighted in yellow the changes within the manuscript.
The revised parts are as follows.
Lines 261, 262, and 263: 
“I2” was changed to “I2.”
Line 682:
"resistant @mutations" has been changed to "resistant mutations."
No revisions were made to the Figures, Tables, or Supplementary information.
Here is a response to the comments of the Reviewer 2.
Comment1: I reviewed the Systematic Review and Network Meta-Analysis entitled “Comparative Efficacy and Safety of Lorlatinib and Alectinib for ALK-Rearrangement Positive Advanced Non-Small Cell Lung Cancer in Asian and Non-Asian Patients”, which is under consideration for publications in Cancers.
Overall, it is a comprehensive review where the author has included data from 1480 studies related to treatment of non-small cell lung cancer. By these analysis, they have compared Lorlatinib to five other treatment options for non-small cell lung cancer, with a special focus on Alectinib.
The review is well balanced and includes treatment outcomes and side effects to the different treatments. The authors has also included different tumor stages and metastasis to the brain, where Lorlatinib shows advantages.
The authors discuss the limitation in a systematic review and limitation in available information, which can have impact on the conclusion. Overall, I find the study ready for publication.
Response1: We wish to express our strong appreciation to the reviewers for their insightful comments on our manuscript.
We are confident that our revised manuscript will be suitable for publication in Cancers and look forward to receiving your editorial decision.
Thank you for your consideration.
Sincerely,
Koichi Ando
Department of Medicine, Division of Respiratory Medicine and Allergology, Showa University School of Medicine
1-5-8 Hatanodai, Shinagawa-ku, Tokyo, 142-8666, Japan
Tel: +81-3-3784-8532
Fax: +81-3-3784-8742
Email: koichi-a@med.showa-u.ac.jp
Reviewer 3 Report
In the current manuscript "Comparative Efficacy and Safety of Lorlatinib and Alectinib for ALK-Rearrangement Positive Advanced Non-Small Cell Lung Cancer in Asian and Non-Asian Patients: A Systematic Review and Network Meta-Analysis" authors investigate and compared safety and efficacy and of several ALK inhibitors in anaplastic lymphoma kinase positive (ALK-p) non-small cell lung cancer (NSCLC) patients. Due to the lack of head-to-head randomized controlled trials (RCTs) they used Bayesian NMA of available datasets to compare the different treatment. They focusing on two compounds lorlatinib and alectinib .
In their network meta-analysis comparison they discovered that some factors such as the race (Asian vs non-Asian) or the presence of CNS metastasis should be taken in account to chose one or the other compound for a better patient response/safety.
Overall the paper is well written and the study is clinically relevant.
I would recommend it for publication.
Minor point: some spell check required i.e. line 682
Author Response
Dear Reviewer 3
Thank you for giving us the opportunity to resubmit to Cancers a revised draft of our manuscript titled “Comparative Efficacy and Safety of Lorlatinib and Alectinib for ALK-Rearrangement Positive Advanced Non-Small Cell Lung Cancer in Asian and Non-Asian Patients: A Systematic Review and Network Meta-Analysis” (Manuscript ID: cancers-1294387). We sincerely appreciate the time and effort that you and the reviewers have dedicated to providing your valuable feedback on our manuscript. We are grateful to the reviewers for their insightful comments. We have made some minor revisions to our manuscript according to the reviewers' comments. We have highlighted in yellow the changes within the manuscript.
Here is a response to the reviewers’ comments and concerns.
Comment1: Some spell check required i.e. line 682.
Response1: Thank you for pointing out this point. We wish to express our strong appreciation to the reviewers for their insightful comments on our manuscript. We have made some minor revisions to some parts of our manuscript including line 682 according to the reviewers' comments.
The revised parts are as follows.
Lines 261, 262, and 263: 
“I2” was changed to “I2.”
Line 682:
"resistant @mutations" has been changed to "resistant mutations."
No revisions were made to the Figures, Tables, or Supplementary information.
We are confident that our revised manuscript will be suitable for publication in Cancers and look forward to receiving your editorial decision.
Thank you for your consideration.
Sincerely,
Koichi Ando
Department of Medicine, Division of Respiratory Medicine and Allergology, Showa University School of Medicine
1-5-8 Hatanodai, Shinagawa-ku, Tokyo, 142-8666, Japan
Tel: +81-3-3784-8532
Fax: +81-3-3784-8742
Email: koichi-a@med.showa-u.ac.jp